# Prospective Observational Study of Prevalence, Assessment and Treatment of Pancreatic Exocrine Insufficiency in Patients with Inoperable Pancreatic Malignancy (PANcreatic Cancer Dietary Assessment—PanDA)

**DOI:** 10.3390/cancers15082277

**Published:** 2023-04-13

**Authors:** Lindsay E. Carnie, Dinakshi Shah, Kate Vaughan, Zainul Abedin Kapacee, Lynne McCallum, Marc Abraham, Alison Backen, Mairéad G. McNamara, Richard A. Hubner, Jorge Barriuso, Loraine Gillespie, Angela Lamarca, Juan W. Valle

**Affiliations:** 1Nutrition & Dietetics, The Christie NHS Foundation Trust, Manchester M20 4BX, UK; 2Department of Medical Oncology, The Christie NHS Foundation Trust, Manchester M20 4BX, UK; 3Division of Cancer Sciences, University of Manchester, Manchester M20 4BX, UK; 4Pancreatic Cancer UK, London SE1 7SP, UK

**Keywords:** pancreatic disease, exocrine insufficiency, gastrointestinal tumours, nutrition support

## Abstract

**Simple Summary:**

In advanced pancreatic cancer, the majority of patients experience pancreatic exocrine insufficiency (PEI), which can negatively impact nutritional status, quality of life and survival rates. Unfortunately, PEI is often diagnosed too late. The aim of our prospective observational study was to identify a screening panel that was simple to use in the outpatient setting, to identify patients that required more timely intervention with pancreatic enzyme replacement therapy. The screening panel proposed is a simple tool that can be used in a clinical setting to identify patients at higher risk of PEI and needing prompt dietetic input.

**Abstract:**

Introduction: Pancreatic exocrine insufficiency (PEI) in patients with advanced pancreatic cancer (aPC) is well documented, but there is no consensus regarding optimal screening. Methods and analysis: Patients diagnosed with aPC referred for palliative therapy were prospectively recruited. A full dietetic assessment (including Mid-Upper Arm Circumference (MUAC), handgrip and stair-climb test), nutritional blood panel, faecal elastase (FE-1) and ^13^C-mixed triglyceride breath tests were performed. Primary objective: prevalence of dietitian-assessed PEI (demographic cohort (De-ch)); design (diagnostic cohort (Di-ch)) and validation (follow-up cohort (Fol-ch)) of a PEI screening tool. Logistic and Cox regressions were used for statistical analysis. Results: Between 1 July 2018 and 30 October 2020, 112 patients were recruited (50 (De-ch), 25 (Di-ch) and 37 (Fol-ch)). Prevalence of PEI (De-ch) was 64.0% (flatus (84.0%), weight loss (84.0%), abdominal discomfort (50.0%) and steatorrhea (48.0%)). The derived PEI screening panel (Di-ch) included FE-1 (normal/missing (0 points); low (1 point)) and MUAC (normal/missing (>percentile 25) (0 points); low (2 points)) and identified patients at high-risk (2–3 total points) of PEI [vs. low-medium risk (0–1 total points)]. When patients from the De-ch and Di-ch were analysed together, those classified by the screening panel as “high-risk” had shorter overall survival (multivariable Hazard Ratio (mHR) 1.86 (95% CI 1.03–3.36); *p*-value 0.040). The screening panel was tested in the Fol-ch; 78.4% patients classified as “high-risk”, of whom 89.6% had dietitian-confirmed PEI. The panel was feasible for use in clinical practice (64.8% patients completed all assessments), with high acceptability (87.5% would repeat it). Most patients (91.3%) recommended dietetic input for all patients with aPC. Conclusions: PEI is present in most patients with aPC; early dietetic input provides a holistic nutritional overview, including, but not limited to, PEI. This proposed screening panel may help to prioritise those at higher risk of PEI, requiring urgent dietitian input. Its prognostic role needs further validation.

## 1. Introduction

Pancreatic ductal adenocarcinoma carries a poor prognosis with a low cure rate, and most patients will die of their disease. Around 41,000 pancreatic cancer-related deaths occurred in Europe in 2014 [1]. Most patients (80%) present with unresectable, advanced pancreatic cancer (aPC); palliative chemotherapy is given, aiming to improve the quality of life (QoL) and prolong overall survival (OS). Around 40% of patients with aPC are not fit for active treatment due to a poor Eastern Cooperative Oncology Group performance status (ECOG-PS) [2].

Patients diagnosed with pancreatic neuroendocrine tumours (PanNETs) differ significantly, and their prognosis is measured in years; they have an estimated median OS of 3.6 years [3] with multiple systemic therapy options available [4]. PanNETs are rare with an estimated incidence of 0.8 per 100,000 [3]. Whilst prognosis is better, a longer survival time means that identifying nutritional deficiencies and minimising their impact is of particular importance [5].

The pancreas produces enzymes to digest carbohydrates, fats and proteins [6]. Tumour location can disrupt this digestive function, causing the development of pancreatic exocrine insufficiency (PEI) [7] and systemic symptoms such as anorexia (83%), asthenia (86%) and weight loss (85%) [8]. Such symptoms impact on nutritional status, QoL [9] and PS, adversely impacting on active treatments [10] including chemotherapy [11].

In pancreatic cancer, PEI is highly prevalent in both resected (>80%) [12] and advanced disease (92%) [13]. Whilst the importance of diagnosing and treating PEI after pancreatic resection is known, it is often overlooked in advanced disease [14].

Early diagnosis of PEI in patients with aPC is important [15,16]. Awaiting the development of PEI-related symptoms such as steatorrhea delays diagnosis and may negatively impact on nutritional status and QoL [17]. There is a lack of consensus for the optimal diagnostic method, as the ^13^C-mixed triglyceride breath test (^13^C-MTBT), which replaced the ‘gold-standard’ three-day faecal fat quantification, is arduous and time consuming [16]. Thus, further investigation of the existing diagnostic methods FE-1 [18] (suggested to be more beneficial in patients not undergoing resection), ^13^C-MTBT [19] and a panel of nutritional markers is warranted [20].

Existing guidelines advise high-dose pancreatic enzyme replacement therapy (PERT) to manage PEI by mimicking ‘normal’ physiological function and normalising nutritional status [19,21,22,23,24]. Additionally, a proton pump inhibitor to increase gastric pH and reduce gastric acid-induced enzyme degradation may enhance the efficacy of PERT [25]. A significant proportion of patients are not prescribed PERT or are prescribed an insufficient dose; available algorithms for dose titration and investigation pathways for refractory symptoms should be used [26]. Although weight loss is a recognised poor prognostic factor in patients with pancreatic cancer [27,28], insufficient research has investigated the extent of nutritionally mediated weight loss, its relation to the cancer and how much could be prevented with pro-active PERT.

This study aimed to define the prevalence of PEI and identify the most appropriate PEI diagnostic panel for use in patients with aPC in clinical practice.

## 2. Methods

For this prospective observational study (NCT03616431), patients diagnosed with aPC (adenocarcinoma, its variants and PanNETs) referred for consideration of palliative systemic therapy were eligible. Advanced pancreatic cancer is defined as a locally advanced disease where surgical intervention is not amenable, or metastatic pancreatic cancer [21]. Consecutive patients were screened for eligibility and written informed consent was gained prior to registration, investigations or assessment. Ethical approval was granted by the North West Greater Manchester East Research Ethics Committee (REC), reference: 17/NW/0597 (IRAS project ID: 194255) [29], with The Christie NHS Foundation Trust Research and the Innovation Department, Manchester, UK acting as study sponsor.

### 2.1. Study Design

The study comprised three separate cohorts, with specific eligibility criteria, clinical assessments, primary and secondary objectives [29,30]. Demographic cohort recruitment opened 1 July 2018 and diagnostic cohort in June 2019. Follow-up cohort recruitment opened upon completion of the interim analysis, allowing the design of the PEI screening panel.

Eligible patients for this prospective observational study were those diagnosed with aPC being considered for palliative chemotherapy, providing written consent to be evaluated by a research dietitian [29]. Specific eligibility, clinical assessment and primary objectives varied between study cohorts (full details in Appendix A). Comorbidities were defined as ‘none’, ‘mild’, ‘moderate’ or ‘severe’ as per the clinician undertaking the new patient consultation.

All patients (demographic and diagnostic cohorts) underwent symptom and full dietetic assessment (including weight, body mass index (BMI), Mid-Upper Arm Circumference (MUAC), handgrip and stair-climb test), full nutritional blood panel including fat-soluble vitamins, faecal elastase (FE-1) and breath test (diagnostic cohort only). Anorexia was defined by FAACT AC/S score and VAS.

Upon completion of the demographic and diagnostic cohorts, the diagnostic panel was determined. This was then prospectively tested in patients within the follow-up cohort.

The primary objectives were: prospective assessment of dietitian-assessed PEI prevalence (demographic cohort), and the design (using breath test as gold standard; diagnostic cohort) and validation (follow-up cohort) of the most suitable screening tool of PEI for patients with aPC.

### 2.2. Statistical Analysis

#### 2.2.1. Planned Sample Size

No formal sample size calculation was performed for any cohort; alternatively, an estimation of possible patient numbers to recruit using established referral rates and the length of study was made. Thus, the study aimed to recruit up to 50 patients completing the “baseline” assessment in the demographic cohort, up to 25 patients willing (at written consent) to complete ^13^C-MTBT and cohort-dependent examinations for the diagnostic cohort, and up to 50 patients completing the “week 4–6” assessments (including a baseline visit and week 4–6 “feedback questionnaire”) in the follow-up cohort. A significant drop-out rate was expected because of poor outcomes associated with aPC; therefore, sufficient patients were recruited to ensure the planned number of ‘evaluable patients’ in each cohort. Non-evaluable patients were replaced until the pre-defined sample size or the end of the study (defined as completion of the three-month follow-up for the last patient recruited into the “follow-up cohort”) were met, whichever happened first.

#### 2.2.2. Data Analysis

Statistical analysis was performed using Stata v12 statistical software.

#### 2.2.3. Descriptive Analyses

Categorical variables were analysed by calculating percentages; continuous variables were summarised using median, range and 95% CI (confidence interval). For exploratory analyses, Student T tests, non-parametric Wilcoxon-Mann-Witney tests, Chi^2^ statistics or Fisher exact tests were utilised, as appropriate.

#### 2.2.4. Design of the Diagnostic Panel

The ^13^C-MTBT test (dichotomised variable (normal (>29) or abnormal (<29)) was considered the “gold-standard” method for diagnosing PEI within the diagnostic cohort. The aim was to select the simplest, most informative panel (from all potential combinations of nutritional bloods, weight, BMI, MUAC, handgrip strength, SC-test, FAACT–A/CS (with VAS), FE-1 and symptom assessment) able to predict the same outcome as the ^13^C-MTBT test. ROC curve analysis was utilised to allow for the dichotomisation of continuous variables if found to be appropriate. The designing of the “screening panel” was performed following three selection steps (full details provided in Appendix A).

#### 2.2.5. Survival Analysis

Survival analyses were performed using the Kaplan–Meier method for the calculation of the estimated median with associated 95% CI. Overall survival was defined as the time from diagnosis of aPC. For comparison of survival curves, Log-rank tests and Multivariable Cox regression analyses were performed.

## 3. Results

Between 1 July 2018 and 30 October 2020, 112 eligible patients were recruited, making up the demographic cohort (50), diagnostic cohort (25), and follow-up cohort (37); patient flow and consort diagrams are presented in Appendix A. Data were last updated on 14 May 2020 and 26 March 2021 for the demographic/diagnostic and follow-up cohorts, respectively.

### 3.1. Prevalence of PEI and Patient Nutritional Status

Baseline characteristics and anti-cancer treatment details for demographic and diagnostic cohorts are provided in Table 1 and were similar between the cohorts (all *p*-values > 0.05), except for a longer median follow-up (*p*-value = 0.014) and more mature data in terms of progression (*p*-value = 0.009) and death (*p*-value < 0.001) events for patients in the demographic cohort (expected due to recruitment timelines).

Full findings following dietitian assessment (including nutritional blood panel) for these two cohorts are provided in Appendix A. In the demographic cohort, the prevalence of PEI based on dietitian assessment was 66.0%. The main PEI-related symptoms were weight loss (84.0%), flatus (84.0%), abdominal discomfort (50.0%) and steatorrhea (48.0%). Anorexia was seen in 74% of patients (defined by FAACT AC/S score and VAS) and 70% of patients required PERT. At baseline, 12% of patients required PERT following dietitian assessment (not already taking it) and 44% and 18% had low levels of vitamins D and A, respectively.

In the diagnostic cohort, PEI prevalence was similar (64%). For the handgrip and SC-tests, worse results were seen in the demographic cohort compared to the diagnostic (*p*-values 0.002 and 0.030, respectively); however, other aspects associated with symptoms, dietetic assessment and nutritional status were similar.

The main findings of bowel changes (type and frequency), prior to diagnosis of aPC and at study entry, are summarised in Appendix A.

Good correlation was seen between the baseline FAACT questionnaire and VAS results to define anorexia when jointly analysed in these two cohorts. Of the 75 patients, 43 and 22 met both anorexia/no anorexia criteria, accounting for an agreement in 65/75 (86.7%) of patients. FAACT was more restrictive, with 9 patients meeting anorexia criteria as per questionnaire, but not as per VAS.

### 3.2. Design of PEI Screening Panel

The requirement to complete the ^13^C-MTBT resulted in a high refusal rate (n = 15). Therefore, the acceptability of the test was assessed on 9 January 2020 after the completion of 10 tests; prior to further recruitment, 8 reported it as acceptable and would undergo again, and thus, recruitment continued.

The ^13^C-MTBT was completed by 19/25 (76%) patients (non-completion reasons included death and study withdrawal) and 16/19 (84%) would undergo the test again. Of those completed, 7 patients (36%) had abnormal results (≤29) and the median result was 31. Using the ^13^C-MTBT to predict PEI diagnosed by dietetic assessment performed poorly (AUC 0.4; 95% CI 0.11–0.67). The acceptability of ^13^C-MTBT was good, 84.2% stating ^13^C-MTBT was “not at all” difficult and 89.5% “not at all” unpleasant. Feedback on FE-1 was slightly worse: 48% stating FE-1 was “not at all” difficult and 40% “not at all” unpleasant; 60% would repeat it.

Utilising the exact logistic regression, the following were identified as clinical variables of potential interest for inclusion in the “allsets” pre-model command analysis: FE-1 (dichotomised; low) [OR 3.85 (*p*-value = 0.5226)], weight loss (dichotomised; yes) [OR 3.98 (*p*-value = 0.4768)], flatus/indigestion (dichotomised; yes) [OR 3.98 (*p*-value = 0.2554)] and MUAC (dichotomised; ≤P25) [OR 14.87 (*p*-value = 0.0348)]. The SC-test (dichotomised; low) was not included due to confounding results (found to be protective factor rather than risk factor; [OR 0.12; *p*-value = 0.0879]) and the number of missing observations (n = 5). Of these, the pre-model “allsets” identified MUAC (dichotomised; ≤P25), FE-1 (dichotomised; low) and flatus/indigestion (dichotomised; yes) as clinical variables of most interest (clinical pre-model characteristics included AUC 0.857, sensitivity 83.3%, specificity 85.7%; *p*-value = 0.038).

Within the blood test findings, vitamin A (dichotomised; low) [OR 7.25; *p*-value = 0.248] and vitamin D (dichotomised; low) [OR 3.27; *p*-value = 0.4339] were preselected as “liposoluble vitamins”; serum total protein (dichotomised; low) [OR 4.67; *p*-value = 0.2456] and prealbumin (continuous variable) [OR 0.01; *p*-value = 0.2804] as “proteins” and transferrin (dichotomised; high) [OR 4.19; *p*-value = 0.3636], international normalised ratio (INR) (dichotomised; high) [OR 0.28; *p*-value = 0.4732] and magnesium (continuous variable) [OR 3.42 × 10^−9^; *p*-value = 0.0739] as “other factors” with exact logistic regression. Following multivariable exact logistic regression within these groups, vitamin A (dichotomised; low) [OR 5.45; *p*-value = 0.3519], serum total protein (dichotomised; low) [OR 3.55; *p*-value = 0.3418] and transferrin (dichotomised; high) [OR 3.01; *p*-value 0.9143] were selected for inclusion in the “allsets” command, together with the clinical variables for identification of the screening panel.

Of the six aforementioned pre-selected factors [MUAC (dichotomised; ≤P25), FE-1 (dichotomised; low), flatus/indigestion (dichotomised; yes), vitamin A (dichotomised; low), serum total protein (dichotomised; low) and transferrin (dichotomised; high)]; the final and most informative (AUC 0.865, sensitivity 83.3% and specificity 87.5%; *p*-value = 0.056) screening panel included MUAC and FE-1. When converting the predictive model into a score, points for each variable/category were assigned proportionally to the OR reported. For this, the OR from the multivariable exact logistic regression included both factors involved in the screening panel: MUAC (dichotomised; ≤P25) [multivariable OR 16.59] and FE-1 (dichotomised; low) [multivariable OR 4.43]. The final screening panel is described in Figure 1. Utilising the point system described, the screening panel had an AUC of 0.8571 (95% CI 0.61367–1.00) to predict ^13^C MTBT-defined PEI; the most informative total point cut-off (ROC curve analysis) was 2, which reached highest sensitivity (85.71%) and specificity (75.00%) and was therefore utilised to dichotomise patients between low-medium risk (0–1 total points) and high risk of PEI (2–3 points).

To validate the design of the screening panel, a sensitivity analysis was performed by repeating the “allsets” command to predict the ^13^C-MTBT results as a continuous variable. This confirmed the selected model with the best performance (lower Akaike information criterion (AIC); reaching an AIC of 16.0 (fifth best out of more than 25) when all six variables were re-tested).

### 3.3. Survival Analysis: PEI and Screening Panel Results

In the joint demographic and diagnostic cohort, OS was explored (follow-up excluded due to limited data maturity). The presence of PEI impacted on OS, in that patients with PEI had a shorter OS: 9.46 months (95% CI 4.47–not reached) vs. 5.26 months (95% CI 3.75–9.07); univariate HR 2.05 (95% CI 1.09–3.83); *p*-value 0.024. In addition, patients classified as “high-risk of PEI” were also reported to have a shorter OS: 10.15 months (95% CI 4.69–14.29) vs. 4.50 months (95% CI 2.27–8.71); univariate HR 2.01 (95% CI 1.41–3.53); *p*-value 0.015. When the multivariable Cox regression model was adjusted for other prognostic factors (type of cancer, ECOG-PS and receipt of palliative chemotherapy), both “high-risk of PEI” and “presence of PEI” were independent prognostic factors associated with shorter OS (multivariable HR of 1.86 (95% CI 1.03–3.36) (*p*-value = 0.040) and HR 2.28 (95% CI 1.19–4.35) (*p*-value = 0.013), respectively) (Figure 2).

### 3.4. Follow-Up Cohort

Baseline characteristics and anti-cancer treatment details are provided in Table 1. Applying the screening panel prospectively to the 37 participants, 24 (64.86%) completed the panel (both MUAC and FE-1), demonstrating the feasibility of applying the panel in clinical practice. Of the 13 patients not fully completing the panel, 12 had missing FE-1 and 1 had both results missing. Of the entire 37-patient cohort, 78.38% (29 patients) were classified as “high-risk” (17 and 12 patients with 2 and 3 points, respectively) and 18.92% (7 patients) classified as low-medium risk (2 and 5 patients with 0 and 1 point, respectively); 1 patient withdrew consent. Comparison of these two cohorts identified that, apart from MUAC and FE-1 included in the screening panel, patients classified as “high-risk” had documented a higher median relative amount of baseline weight loss (18.5% vs. 5.3%; *p*-value 0.041); baseline weight (median 66 kg vs. 77.25 kg; *p*-value 0.0458) and BMI (median 22.96 vs. 28.03; *p*-value 0.0001) were also lower (Appendix A). No other significant differences were identified. Of these three factors, multivariable logistic regression identified a higher BMI (≥25), to be associated with lower chance of a “high-risk” result (multivariable HR 0.04 (95% CI 0.01–0.81); *p*-value 0.036); therefore, patients with a low BMI could be prioritised for the screening panel. The impact of the screening panel on OS could not be assessed due to limited follow-up in this cohort (40.54% of death events; median follow-up 2.49 months (0–13.21) (1.75–3.27)).

A screening panel acceptability questionnaire was completed by 24 patients; 91.67% finding it “not at all” difficult and 87.50% “not at all” unpleasant; 87.50% would repeat. Most patients (91.3%) provided feedback about the dietitian, recommending dietetic input for all future patients diagnosed with aPC (Table 2). Most patients considered dietetic intervention as “very much” helpful (82.61%), that it improved QoL “very much” (30.43%) or “significantly” (43.48%). Symptom improvement varied: “very much” (26.09%), “significantly” (30.43%) and “not too much” (26.09%).

Comparison of the screening panel and dietitian was assessable in 29 patients; agreement was 89.7% (26/29), dietitian confirming PEI in 20/23 patients with a “high-risk” panel result (agreement 87.0%); agreement maintained at 6-week (86.0%) and 3-month (83.0%) visits. Reasons for disagreement (baseline visit; n = 3) included: low MUAC due to poor nutritional intake secondary to pain and stress (n = 1), no symptoms and normal FE-1 despite high-risk panel (n = 1) and symptoms difficult to assess due to newly formed ileostomy (n = 1). Due to the low number of patients without PEI in this cohort, the agreement between a “low-risk” result and the dietitian (“no PEI”) was limited: at baseline, no patients classified as “low-risk” had “no PEI” as per dietitian assessment; 33.3% at week 6 and 50.0% at month 3.

Compliance of PERT was 80.0% at week 6 and 90.9% at month 3 visits. PERT-associated AEs included itchy skin in 1 patient (reported at the week 6 follow-up visit).

During follow-up visits (Appendix A), only one patient was newly diagnosed with PEI (new PERT started) at week 6; none at month 3. However, dietetic input for PERT dose adjustment was required for 3 patients at the month 3 follow-up, suggesting that ongoing dietetic support is required. Compliance of screening panel completion at follow-up visits reduced significantly (40.5% at week 6, 13.5% at month 3). Based on longitudinal screening panel results (Appendix A), a worsening result was seen at week 6 in 4/7 patients (57.1%) who were low risk at baseline. No patients classified as low risk at week 6 were deemed high-risk at month 3. Therefore, repeating the panel at week 6 could be useful, whilst the benefit at month 3 is debatable. An improvement in the screening panel result was only seen in 1/29 (3.5%) at week 6, and another 1/16 at month 3 (6.3%) of those with a “high-risk” result at the previous visit. Three patients had newly identified low vitamin D levels during a follow-up visit: not the case for vitamin A. This supports repeating vitamin D testing during follow-up, if the prior level was normal, whilst measuring other nutritional markers beyond baseline is of little use.

Data for QoL at baseline and changes over time were assessed; questionnaires were completed by 86.5%, 48.7% and 35.1% of patients at baseline, week 6 and month 3 visits, respectively (Appendix A). In terms of QoL, global health was poor (Appendix A), with pain (including pancreatic pain) and insomnia being symptoms that improved the most over time (Appendix A) for patients with pancreatic ductal adenocarcinoma.

Based on these results, the following are recommended (summarised in Figure 3).

## 4. Discussion

Most patients with aPC have PEI. As PEI can negatively impact QoL and prognosis, an early holistic nutritional assessment, including diagnosing and managing PEI by a dietitian is required.

Whilst research has proposed a role for testing nutritional markers, testing FE-1 and utilising ^13^C-MTBT to identify PEI in this patient cohort, there continues to be a lack of consensus on the most optimal method. A screening panel could aid in detecting those at higher risk of developing PEI.

As we are unable to identify those at higher risk using weight/BMI alone, this proposed screening panel should be completed for everyone at diagnosis. However, patients with a low BMI could be prioritised. Screening at baseline/diagnosis and at week 6 review is beneficial but further repetition may be of little use. Whilst the panel is good at identifying PEI, it is not as good at excluding PEI.

There is a proven benefit of monitoring vitamin D levels throughout the patient pathway, as low levels were seen at later reviews.

Whilst the importance of PERT is recognised, and most patients were prescribed PERT prior to baseline, 46% and 48% patients in the demographic and diagnostic cohorts, respectively, required dose adjustment, with ongoing symptoms in-keeping with PEI. A small number of patients deemed high-risk of PEI were not taking PERT. Whilst few patients were diagnosed with PEI at the later reviews, some required PERT dose adjustments, supporting ongoing dietetic input.

Despite patients considering the ^13^C-MTBT acceptable, 15 patients declined study participation due to needing to undertake this test. One could argue that in this patient cohort, it is not a suitable test in clinical practice. On the contrary, this screening panel was proven as acceptable and feasible in an outpatient setting. Whilst feasible, some patients were not well enough to complete, supporting the argument for early intervention.

Unfortunately, many FE-1 were not completed and requesting repeat samples was challenging. MUAC alone by a trained clinician is the minimum requirement as FE-1 is of much more importance if MUAC is normal. Missing MUAC and FE-1 results meant only a limited number of patients completed the screening panel. Requesting FE-1 testing at diagnosis, alongside other tests, might facilitate a higher completion rate and earlier PEI diagnosis.

Anorexia is prevalent in aPC and should be assessed and addressed early; it may minimise weight loss and improve QoL. Good correlation was seen between FAACT A/CS and VAS tools to define anorexia. As VAS is more restrictive, a combination of both tools is probably of most interest.

Whilst it could be argued that all patients with aPC should be prescribed PERT, dietetic input remains imperative to monitor and dose adjust as required. Input was deemed important to patients within this study. Differences in the handgrip and SC-test results between demographic and diagnostic cohorts were likely due to selection bias as other aspects of dietetic assessment were similar.

Strengths of this study: involvement of a multidisciplinary team ensuring a holistic approach to the research and patient care. Dietetic input was provided for all patients involved in the study, ensuring gold-standard care and early intervention for dietetic and nutritional issues. Additionally, a screening panel that is simple and can be translated to different outpatient settings has been designed.

Limitations also exist; small sample sizes, difficulty recruiting to the diagnostic cohort due to the ^13^C-MTBT and poor PS. Unfortunately, a large proportion of FE-1 tests were not completed/reported; no sample was provided, or services were unavailable during the COVID-19 pandemic. Dietetic reviews were by phone due to the COVID-19 pandemic and hospital policy changes may have impacted on compliance with advice. Limited follow-up of patients meant that the impact of the screening panel on OS could not be assessed. Patients diagnosed with both PanNETs and PDAC were included in this study, which could cause bias as the rate of PEI occurring in these cohorts does differ. Another limitation of the study is the lack of health economic assessment to explore the cost benefit impact of the treatment of PEI in this setting.

## 5. Conclusions

Early diagnosis and treatment of PEI in aPC is of upmost importance. The proposed screening panel could help to identify and prioritise those at higher risk of PEI, requiring more timely dietetic input. Despite most patients experiencing PEI, dietitians play an important role in providing individualised treatment to titrate the PERT dose as required. The prognostic role of this screening panel requires further validation.

## Figures and Tables

**Figure 1 cancers-15-02277-f001:**
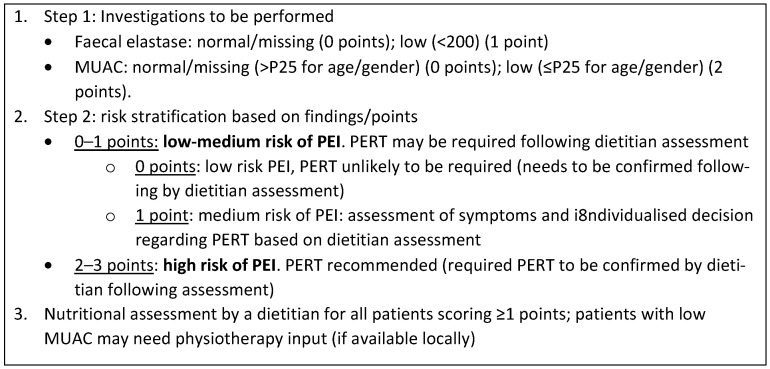
Screening panel for pancreatic exocrine insufficiency. Abbreviations: MUAC, mid-upper arm circumference; PEI, pancreatic exocrine insufficiency; PERT, pancreatic enzyme replacement therapy.

**Figure 2 cancers-15-02277-f002:**
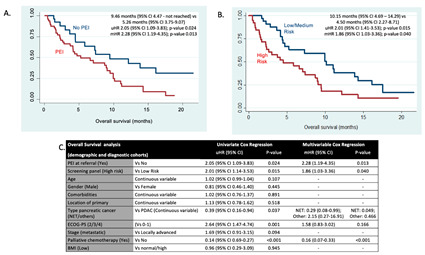
Joint survival analysis for demographic and diagnostic cohorts: both PEI and the screening panel were prognostic factors. Abbreviations: CI, confidence interval; ECOG-PS, Eastern Cooperative Oncology Group Performance Status; mHR, multivariable hazard ratio; NET, neuroendocrine tumour; PDAC, pancreatic ductal adenocarcinoma; PEI, pancreatic exocrine insufficiency; uHR, univariate hazard ratio. (**A**) Kaplan-Meier curve for joint demographic and diagnostic cohorts comparing PEI vs. no PEI. (**B**) Kaplan-Meier cure for joint demographic and diagnostic cohorts comparing Low/Medium risk screening panel vs. High-risk screening panel (**C**) Overall survival analysis adjusting for other prognostic factors.

**Figure 3 cancers-15-02277-f003:**
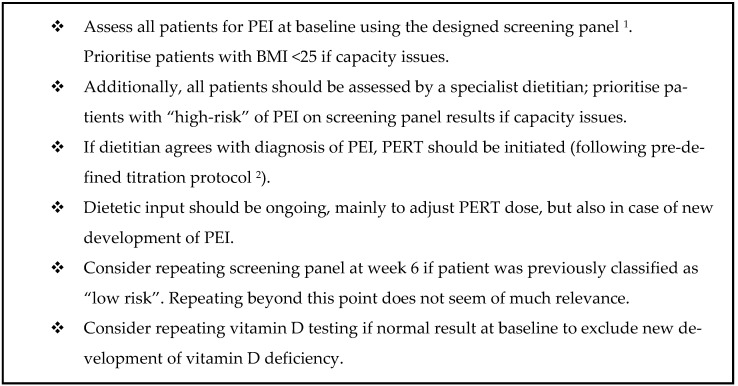
Recommendations. Abbreviations: BMI, body mass index; PEI, pancreatic exocrine insufficiency, PERT, pancreatic enzyme replacement therapy. ^1^ in Table 2. ^2^ in Appendix A.

**Table 1 cancers-15-02277-t001:** Baseline characteristics and details on treatment administered and patient outcomes. To compare the demographic and diagnostic cohorts, T-Tests and Chi square of Fish were utilised (excluding missing values). Abbreviations: CisGem, cisplatin Capecitabine; CI, confidence interval; ECOG PS, Eastern Cooperative Oncology Group Performance Status; FOLFOX, Foloni acid, fluorouracil and oxaliplatin; GemCap, gemcitabine capecitabine; GemNabPaclitaxel, Gemcitabine Nab-paclitaxel; Ki67, nuclear protein 67; NET, neuroendocrine tumour; SSA, somatostatin analogue; TemCap, Temozolomide-capecitabine. * other types of chemotherapy combinations.

		Demographic Cohort (n = 50)		Diagnostic Cohort (n = 25)		Demographic vs. Diagnostic Cohort	Joint Demographic and Diagnostic Cohorts (n = 75)		Full Follow-Up Cohort (n = 37)	
		n	%	n	%	*p*-Value	n	%	n	%
Baseline Characteristics										
Age at study entry	Median (range) (95% CI)	65.63 (25.56–87.53) (60–68)		69.63 (50.98–84.68) (64–72)		0.2259	65.87 (25.56–87.53) (62–68)		70.57 (44–85.11) (65–72)	
Gender	Female	27	54.0	11	44.0	0.414	38	50.67	17	45.95
	Male	23	46.0	14	56.0		37	49.33	20	54.05
Comorbidities	None	20	40.0	17	68.0	0.116	37	49.33	13	35.14
	Mild	19	38.0	4	16.0		23	30.67	18	48.65
	Moderate	7	14.0	2	8.0		9	12.0	5	13.51
	Severe	4	8.0	2	8.0		6	8.0	1	2.7
Localisation primary pancreatic tumour	Head/neck	25	50.0	13	52.0	0.713	38	50.67	20	54.05
	Body	16	32.0	6	24.0		22	29.33	10	27.03
	Tail	9	18.0	6	24.0		15	20.0	6	16.22
Biopsy confirmed cancer	No	1	2.0	0	0.0	1.000	1	1.33	0	0.0
	Yes	49	98.0	25	100.0		74	98.67	37	100.0
Type of pancreatic cancer	Adenocarcinoma	44	88.0	20	80.0	0.064	64	85.33	33	89.19
	NET	6	12.0	2	8.0		8	10.67	2	5.41
	Other	0	0.0	3	12.0		3	4.0	2	5.41
Differentiation (if NET)	Grade 1	1	2.0	0	0.0	1.000	1	1.33	**1**	**-**
	Grade 2	3	6.0	1	4.0		4	5.33	0	-
	Grade 3	2	4.0	1	4.0		3	4.0	1	-
	Not NET	44	88.0	23	92.0		67	89.33	35	-
Ki 67 (if NET)	Median (range) (ki67)	10 (2–80) (0–56.45)		17.5 (3–32) (0–100)		0.7777	10 (2–80) (0.12–45.38)		14 (1–27) (0–100)	
Functional (if NET)	Yes	0	0.0	0	0.0	n/a	0	0.0	0	-
	No	6	12.0	2	8.0		8	10.67	2	-
	Not NET	44	88.0	23	92.0		67	89.33	35	-
ECOG PS at study entry	0	7	14.0	9	36.0	0.230	16	21.33	6	16.22
	1	28	56.0	11	44.0		39	52.0	21	56.76
	2	12	24.0	5	20.0		17	22.67	7	18.92
	3	2	4.0	0	0.0		2	2.67	3	8.11
	4	1	2.0	0	0.0		1	1.33	0	0.0
Stage at study entry	Localised	0	-	0	-		0	-	1	2.7
	Locally advanced	16	32.0	7	28.0	0.723	23	30.67	18	48.65
	Metastatic	34	68.0	18	72.0		52	69.33	18	48.65
Treatment and outcomes										
Treatment intention	Palliative	50	100.0	25	100.0	n/a	75	100.0	37	100.0
Did patient received systemic treatment	No	13	26.0	4	16.0	0.386	17	22.67	11	29.73
	Yes	37	74.0	21	84.0		58	77.34	26	70.27
Line of treatment	First-line	35	70.0	21	84.0	0.263	56	74.67	25	67.57
	Other line	2	4.0	0	0.0		2	2.67	1	2.7
	None	13	26.0	4	16.0		17	22.67	11	29.73
Type of systemic treatment	Gemcitabine	6	12.0	3	12.0	0.834	9	12.0	7	18.92
	FOLFIRINOX	9	18.0	6	24.0		15	20.0	5	13.51
	GemCap	5	10.0	4	16.0		9	12.0	7	18.92
	GemNabPaclitaxel	6	12.0	4	16.0		10	13.33	5	13.51
	Sunitinib	0	0.0	1	4.0		1	1.33	0	0.0
	SSA	2	4.0	0	0.0		2	2.67	1	2.7
	TemCap	3	6.0	0	0.0		3	4.0	0	0.0
	Carboplatin and Etoposide	1	2.00	1	4.0		2	2.67	0	0.0
	Other *	5	10.0	2	8.0		7	9.33	1	2.7
	None	13	26.0	4	16.0		17	22.67	11	29.73
* If Other (which)	NUC1031	2	-	0	-	n/a	2	-	0	-
	CisGem	0	-	1	-		1	-	1	-
	FOLFOX + NabPaclitaxel	3	-	1	-		4	-	0	-
Chemotherapy dose intensity (%)	Median (range) (95% CI)	68.5 (11.1–100) (58.96–77.37)		80.6 (11.1–100) (55.06–84.28)		0.8514	73.9 (11.1–100) (61.10–76.28); 54 observations		52.6 (11.1–100) (40.88–58.84)	
Best radiological response	Progressive disease	4	8.0	2	8.0	0.350	6	8.0	3	8.11
	Stable disease	12	24.0	8	32.0		20	26.67	12	32.43
	Partial response	14	28.0	3	12.0		17	22.67	4	10.81
	Not documented or no treatment received	20 (13 never started treatment)	40.0	12	48.0		32 (17 never started treatment)	42.67	18	48.65
Radiological progression documented at time of last data lock	Yes	22	44.0	5	20.0	0.009	27	36.0	5	13.51
Death documented at time of last data lock	Yes	41	82.0	10	40.0	<0.001	51	68.0	15	40.54
Overall survival (estimated)	Median (95% CI)	7.39 (1.14–9.95)		5.85 (4.11-nr)		0.79784 (log rank)	7.29 (4.37–9.49)		4.27 (2.17–nr)	
Follow-up	Median (range) (95% CI)	7.23 (0.16–21.59)		3.28 (0.46–9.03)		0.014	4.50 (0.16–21.55) (5.39–8.01)		2.49 (0–13.21) (1.75–3.27)	

**Table 2 cancers-15-02277-t002:** Patient feedback of dietetic input in the follow-up cohort.

		Follow-Up Cohort (n = 37)	
		N	%
Dietitian feedback questionnaire completed (n = 37)	No	14	37.84
	Yes	23	62.16
Test completed (if questionnaire done)	No	0	0.00
	Yes	23	100.00
Dietitian intervention was helpful (if questionnaire done)	1 (≥)	19	82.61
	2 (significantly)	4	17.39
	3 (not too much)	0	0.00
	4 (not at all)	0	0.00
	Not answered	0	0.00
Dietitian intervention improved my quality of life (if questionnaire done)	1 (very much)	7	30.43
	2 (significantly)	10	43.48
	3 (not too much)	2	8.70
	4 (not at all)	2	8.70
	Not answered	2	8.70
Dietitian intervention improved my symptoms (if questionnaire done)	1 (very much)	6	26.09
	2 (significantly)	7	30.43
	3 (not too much)	6	26.09
	4 (not at all)	2	8.70
	Not answered	2	8.70
I would suggest the dietitian input to be available for other patients in the future	Yes	21	91.30
	No	1	4.35
	Not answered	1	4.35
Free text comments	I have only spoken with the dietician once, however I have listened to her advice and this has enabled me to gain 2 kg After the info and help from the dietician, I was able to carry on as normal a life, as possible, knowing that I had got enough info to get everything right for now Trying to put weight on is not easy as I am having to try to eat much more than I would have, balancing diet with Creon seems to be a matter of experience and fine tuning Very helpful and informative and feel it should be available to all Have put on weight. Critical to have input from dietician.		

## Data Availability

The data presented in this study is contained within the article or Appendix A.

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
