# Peer review of "Prospective Observational Study of Prevalence, Assessment and Treatment of Pancreatic Exocrine Insufficiency in Patients with Inoperable Pancreatic Malignancy (PANcreatic Cancer Dietary Assessment—PanDA)"

_cancers, 2023, doi:10.3390/cancers15082277_

Round 1
Reviewer 1 Report
DEAR AUTHORS
Your manuscript is very interesting. The topic is about PEI. The treatment improvement about PEI is necessary and your manuscript could help
My comments: Abstract and introduction: OK. Methods. I think that every physician involved in pancreatic cancer should define correctly advanced pancreatic cancer but in Methods section is necessary to define the term. NCCN guidelines or other classification? Please include your definition of aPC in Methods section. I do not agree mixing NET and adenocarcinoma as you explain in Introduction the rate of IPe is not the same and the cases of NET are very few I am not going to ask to do all statistician analysis without NET cases but I think that could cause some bias. The rates of IPE in tail and head cancers are different in the literature (I am not sure about these data from literature). You should comment as limitation.
Results: The not confirmed biopsy case should be excluded. In table 1 please erase the line about localized cases is confusing. You have 0 cases (logical is a exclusion criteria but in follow up there is one case: downsizing? In results you talk about Anorexia definition and this point should be in Methods section. In results you talk about vitamins but you did not mention in Methods. Information about how you did your model is Methods. Results section is too long and sometimes difficult to follow. A multivariate analysis checking that survival is related to IPE but excluding tumors load (metastatic vs only advanced is mandatory).
Discussion is very short and without comparison with previous literature
You include limitations but please mention strengths
Author Response
Dear reviewer,
We thank you for your comments and suggestions. Please see the attachment for the responses.

Reviewer 2 Report
The manuscript highlighting the importance of evaluating pancreatic exocrine insufficiency in patients advanced pancreatic cancer is well and elaborately written. The manuscript is of interest. The conclusion is supported by the data and the authors have included the limitations of the study. The following concerns should be addressed.
Please include what comorbidities were present in the population under study.
All gradings used in supplementary material should be defined in the legends-
supp mat 7, 8, 9: PEI grades; steatorrhea grades, flatus indigestion, abdominal discomfort, bowel movement types, etc.
Please discuss the translational significance and cost benefit ratio in the discussion section or in limitations.
Author Response
Dear reviewer,
We thank you for your comments and suggestions. Please see the attachment for the responses and changes made to the manuscript.

Round 2
Reviewer 1 Report
Dear Authors
Manuscript has improved with your changes